# Sex differences in the physiological response to acute anterior cruciate ligament overuse

Stephen H. Schlecht[1,2] , Benjamin E. Loflin[1], Adam R. Carter[1], Roufael Hanna[3], Simran Shergill[4] and Edward M. Wojtys[5]

[1]*Department of Orthopaedic Surgery, Indiana University School of Medicine, Indianapolis, IN, USA*
[2]*Department of Anatomy, Cell Biology and Physiology, Indiana University School of Medicine, Indianapolis, IN, USA*
[3]*Department of Kinesiology, Indiana University – Indianapolis, Indianapolis, IN, USA*
[4]*Department of Biology, Indiana University – Indianapolis, Indianapolis, IN, USA*
[5]*Department of Orthopaedic Surgery, University of Michigan, Ann Arbor, MI, USA*

Handling Editors: Paul Greenhaff & Martino Franchi

The peer review history is available in the Supporting Information section of this article (https://doi.org/10.1113/JP289078#support-information-section).

**The Journal of Physiology**

Stephen Schlecht, an Assistant Professor of Orthopaedic Surgery at Indiana University School of Medicine, specializes in musculoskeletal mechanobiology. His basic, translational and clinical research focuses on elucidating how ligaments structurally and functionally respond to activity-driven matrix damage and restore tissue integrity *in vivo*. The overarching goals of this research are to develop new clinical diagnostics for ligamentous injury prevention in young athletes, and to improve post-surgical outcomes to mitigate the risk of developing debilitating joint pathologies such as arthrofibrosis and post-traumatic osteoarthritis.

**Abstract figure legend** With equivalent *in vivo* fatigue loading of the anterior cruciate ligament (ACL), females accumulate more tissue damage and mount a slower reparative response than males, when the applied loads between sexes are equivalent. By contrast, males accumulate less tissue damage and mount a robust reparative response that includes resolving inflammation and promoting cell proliferation and migration. With translation, this differential physiological response to ACL overuse may, in part, explain higher non-contact injury rates in females.

**Abstract** Young female athletes are at least two times more likely to suffer a non-contact anterior cruciate ligament (ACL) injury than males, and one and a half times more likely to have a recurrent injury. Primary factors contributing to this disparity are less stiff and weaker ACLs, and greater knee laxity than males. Also, some evidence suggests females may exhibit a muted response to repetitive, high-intensity activity compared to males. Here, we test the hypothesis that female ACLs would accumulate more extracellular matrix (ECM) damage and show a delayed reparative response compared to males under equivalent submaximal fatigue loading. Using an adolescent mouse model (C57BL/6J), ACLs were cyclically loaded to induce an acute submaximal overuse injury ($n = 20$ per sex). ECM damage was assessed via immunofluorescence, apoptotic activity via immuno-histochemistry, and gene expression changes through RNA-sequencing at 24 and 72 h post-injury. Female ACLs showed significantly greater collagen denaturation than males ($P = 0.05$), with no significant difference in apoptosis. Transcriptomic analyses suggest sex-specific healing strategies. Females followed a slower, more regulated reparative response, whereas males exhibited a more aggressive repair approach emphasizing mitosis, cell proliferation and migration. These findings may explain higher female ACL failure rates as greater matrix damage combined with a slower repair response could lead to injury propagation if reloading occurs prematurely. By contrast, the faster male response might reduce recurrence risk but increase fibrosis potential. If confirmed further, these potential physiological differences may require the implementation of sex-specific strategies for training and recovery regimens to prevent overuse injuries and optimize healing outcomes in young athletes.

(Received 29 April 2025; accepted after revision 23 September 2025; first published online 29 October 2025)

**Corresponding author** Stephen H. Schlecht: Indiana University School of Medicine, VanNuys Medical Science Building, Room 549, 635 Barnhill Drive, Indianapolis, IN 46202, USA.    Email: steschle@iu.edu

## Key points

- Females are at least twice as likely to suffer an anterior cruciate ligament injury (ACL) relative to males when participating in the same, or comparable sport.
- Sex differences in ACL structure and function are primary factors for increased female injury risk.
- Here, we show that the female mouse ACL accrues more collagen matrix damage than males under comparable fatigue loads.
- Additionally, female mice demonstrated a slower, more regulated ACL reparative response, while male mice exhibited a more aggressive repair approach emphasizing mitosis, cell proliferation and migration.
- These results, if translatable, may in part explain the sex-disparity in ACL failure rates with greater matrix damage and a slower repair response increasing the risk of reinjury, and a faster male response potentially reducing injury recurrence risk with further physical activity.

## Introduction

Anterior cruciate ligament (ACL) injuries are one of the most common orthopaedic injuries in the USA, with at least 200,000 cases annually (Sanders et al., 2016). More than half of ACL injuries occur in individuals between 15 and 25 years of age (Griffin et al., 2006) and primarily result from non-contact mechanisms (Boden et al., 2000). Among high school and collegiate athletes, females are at least two times more likely to suffer a first-time non-contact ACL failure than males when accounting for sport played (Beynnon et al., 2014) and at

least one and a half times more likely to suffer a recurrent ACL injury (Slater et al., 2019). ACL injuries can be debilitating and increase the likelihood of time away from sport, discontinuation of sport and surgical intervention (Swenson et al., 2013; Welton et al., 2018). Moreover, those that suffer an ACL injury are four to six times more likely to develop knee osteoarthritis (Poulsen et al., 2019), irrespective of receiving conservative or surgical treatment (Friel & Chu, 2013; Roos, 2005).

Reasons for this sex disparity in injury risk are multifactorial and include a suite of anatomical, biomechanical and hormonal differences (Ireland, 2002; Shultz & Fegley, 2023; Shultz et al., 2008). Importantly, females on average have less stiff and weaker ACLs compared to males (Chandrashekar et al., 2006; Lipps et al., 2012), contributing to a greater knee laxity (Shultz et al., 2008). Fluctuations in the female hormonal milieu during the peri-ovulatory and mid-luteal days of the menstrual cycle are associated with increased knee laxity and greater anterior tibial translation upon weight acceptance (Herzberg et al., 2017; Shultz et al., 2011; Somerson et al., 2019). Moreover, studies using ovariectomized female animals have shown ACL mechanical properties decrease with oestrogen supplementation, although at supraphysiological levels (Slauterbeck et al., 1999). Furthermore, clinical findings suggest that these soft tissue functional changes arise via sex hormone modulation of collagen metabolism (Shultz et al., 2012). Recent *in vitro* findings from cyclically fatigued rabbit ACLs support this perspective, with female ACLs demonstrating a muted expression of pro-collagen synthesis genes, but greater upregulation of catabolic genes, whereas male ACLs showed significant upregulation of pro-collagen genes and little change in catabolic gene expression within the extracellular matrix (ECM) (Paschall et al., 2024). Thus, the collagen remodelling response to loading may be delayed, or muted, in female ACLs, potentially increasing their risk for failure with subsequent physical activity. This is an important finding considering data increasingly suggest that the accumulation of load-induced ECM damage via repetitive high-intensity activity probably precedes many non-contact ACL injuries (Chen et al., 2019; Kim et al., 2022; Loflin et al., 2023; Putera et al., 2023).

Here, we sought to further investigate these sex-based metabolic differences in the response of the ACL to cyclic loading (i.e. fatigue) *in vivo*, at the same time as determining whether female ACLs demonstrate a greater propensity for collagen matrix damage via acute tissue overuse. We hypothesized that female ACLs would accrue significantly more matrix damage and exhibit a delayed, or prolonged, response to collagen disruptions relative to male ACLs fatigued at equivalent loads. To test this hypothesis, we employed a novel mouse ACL fatigue model to generate an acute over-

use injury *in vivo* that we previously showed induces collagen matrix denaturation (i.e. damage) and reduced organ-level ACL strength and stiffness (Loflin et al., 2023). We then used histological and molecular assays to characterize ECM structural alterations (collagen denaturation, programmed cell death) and the cellular response (differential gene expression) to an acute overuse injury.

## Methods

### Ethical approval

All animal experiments were approved by the Indiana University School of Medicine Institutional Animal Care and Use Committee, which adheres to Association for Assessment and Accreditation of Laboratory Animal Care International guidelines. The institutional approval code for the protocol detailing animal care, experimentation and humane death is No. 22062.

### Experimental design

Male and female C57BL/6J inbred mice were ordered from The Jackson Laboratory at 9 weeks of age ($n = 20$ per sex). Mice were group housed by sex under a 12:12 h light/dark photocycle cycle and provided food and water *ad libitum*. Prior to beginning the study, mice were allowed to acclimate to their new environment for 1 week. At 10 weeks of age, which is an age musculoskeletally comparable to adolescent humans (Dutta & Sengupta, 2016), mice were provided heat support, anesthetized using vaporized isoflurane concentrations appropriate for their weight (5% induction, 1.5–2% maintenance) and had ocular lubrication applied. Anaesthesia depth was monitored via heartrate and paw withdrawal reflex. When sedated, each mouse had their right knee positioned in a custom loading fixture that approximates knee kinematics involved in a jump-landing and pivot shift to cyclically load the ACL to induce an acute overuse injury. Mice were then recovered from anaesthesia with the provision of heat support, monitored for 1 h and returned to normal cage activity with no analgesia provided. Mouse recovery is monitored every 12 h for up to 3 days following experimentation. Twenty-four hours after injury, a subset of male and female mice ($n = 10$ per sex) were randomly selected and killed via isoflurane overexposure followed by cervical dislocation for histological and molecular analyses. The remaining mice ($n = 10$ per sex) were killed in the same manner 72 h following injury for the same downstream analyses. Immediately following death, paired hindlimbs were removed from each mouse. The ACLs of four paired hindlimbs were processed for histology. The remaining six paired hindlimbs from each

post-injury timepoint were dissected from the knees and processed for RNA sequencing.

### *In vivo* ACL fatigue loading and knee mechanics

The *in vivo* mouse model accounts for jump-landing/pivot shift knee kinematics previously employed to characterize the ACL fatigue mechanism *in situ* using cadaveric human knees (Chen et al., 2019; Kim et al., 2022; Putera et al., 2023). For mice, a custom loading jig aligns the knee joint with the testing system actuator at the same time as statically applying a valgus moment across the knee joint and internally rotating the tibia, as previously described (Fig. 1) (Loflin et al., 2023). To fatigue the ACL, mice were provided heat support, anesthetized isoflurane and ocular lubrication. Next, mice were positioned in the loading fixture with a preload of ∼0.25 N applied to the right distal femur to maintain femoral–tibial contact within the knee throughout the loading protocol to prevent ACL trauma confounders (e.g. bone bruising, subchondral fracture) during fatiguing. Right knees were then exposed to interval loading (moderate to strenuous cyclic loading) that mimics load intensity shifts experienced during athletic training and competition, and was previously shown to generate increased *in vivo* collagen matrix denaturation accompanied by a progressive reduction in ACL strength and stiffness (Loflin et al., 2023). The left knees served as internal controls. To define the moderate and strenuous loads prior to testing, a subset of mice ($n = 10$ per sex) had their right ACLs loaded to failure using the exact same fixture and testing protocols as for fatigue loading, except that, instead of cycling the load, the distal femur was axially loaded at a rate of 2.7 N s$^{-1}$ until an audible 'pop' or a 10% decrease in load occurred, indicating the maximum load to failure. Mice were then recovered, returned to vivarium and monitored for use in an unrelated study. The failure model generates partial ACL tears within the proximal third with no macropathology to the surrounding joint structures (Ahn et al., 2023). From the resulting load-displacement curves generated from each failure, moderate and strenuous submaximal loads were defined as 30% and 60%, respectively, of the body weight adjusted median *in vivo* ACL maximum load to failure. After determining the prescribed loads for each sex, the right knee and ACL were fatigue loaded for 440 cycles at a rate of 0.75 mm s$^{-1}$. Over the course of the fatiguing protocol, the load was programmed to routinely shift between moderate and strenuous loading. The loading protocol began with 20 cycles at 30% max ACL failure load followed by 10 cycles at 60% maximum ACL failure load, which was then repeated throughout the test. After training, mice were recovered with heat support, returned to vivarium and monitored until death.

From the load-displacement data acquired during fatigue loading of each mouse, a custom MATLAB code (MathWorks Inc., Natick, MA, USA) was utilized to calculate knee hysteresis and stiffness beginning at the 51st cycle to ensure that the knee mechanics had stabilized (Fung et al., 2010). Hysteresis was quantified as the area between the loading and unloading curves, which is a measure of the knees viscous response (Maganaris & Paul, 2000). Knee stiffness was quantified as the average of slopes of the linear portion of the loading and unloading curves, which is a measure of anterior knee laxity (Loflin et al., 2023). Hysteresis and stiffness were calculated separately for the 30% and 60% load/unload curves.

### Histochemistry

At 24 and 72 h post-fatigue, a subset of paired hindlimbs were processed for paraffin histology ($n = 4$ per sex per timepoint). Immediately following death, paired hindlimbs were removed and fixed in 4% paraformaldehyde for 48 h, followed by rinsing in distilled water. Tissue was then decalcified in 10% ethylenediamine tetraacetic acid at 4°C until decalcification was confirmed via radiographic assessment (∼10 days), followed by rinsing in distilled water. Finally, tissue was serially dehydrated through graded ethanol and embedded in paraffin. Paraffin-embedded tissue blocks of each knee were sectioned sagittally at 5 μm and encompassed the entire expanse of the ACL. Following this, a subset of tissue sections from mice killed 24 h post-injury were deparaffinized and incubated with a sulfo-Cyanine3 fluorophore-conjugated collagen hybridizing peptide (R-CHP; 3Helix, Salt Laek City, UT, USA). The CHP is specific to denatured collagen of all types (Zitnay et al., 2017). Additionally, tissue sections from mice killed at both post-injury timepoints were incubated with a cleaved caspase-3 primary antibody and a biotinylated

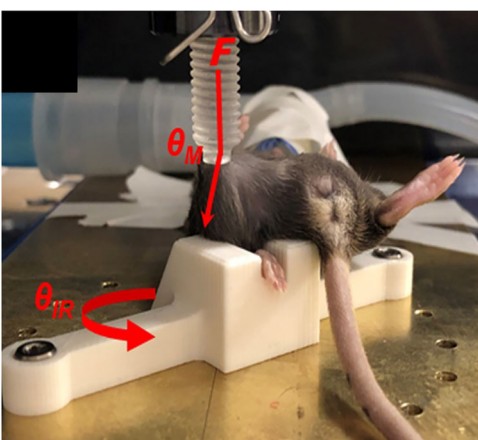

**Figure 1. *In vivo* testing setup**
Knee in valgus, tibia internally rotated. Cyclic (fatigue) or maximal (failure) compressive force applied to femur to tension ACL.

goat anti-human IgG secondary antibody (Vectastain Elite ABC-HRP Kit; Vector Laboratories, Newark, CA, USA) for detection of apoptotic cells within the ECM using 3,3′-diaminobenzidine (ImmPACT DAB; Vector Laboratories). Stained images were acquired at 10× magnification under fluorescence (TRITC filter, 250 ms exposure, 1× gain) or brightfield using a motorized microscope (Eclipse Ni-U; Nikon Corp., Tokyo, Japan) equipped with a 16.25-MP monochrome camera (DS-Qi2; Nikon Corp.). The percentage positive CHP and caspase-3 area was quantified across the entire ACL and associated entheses. For CHP, fluorophore expression was quantified using ImageJ (National Institutes of Health, Bethesda, MD, USA) and a custom macro to calculate the percentage area of ACL tissue containing denatured collagen. For caspase-3, quantification of DAB stain was performed using ImageJ via colour thresholding and the Immuno-histochemistry Image Analysis Toolbox plug-in (Shu et al., 2013), to calculate the percentage area of tissue containing apoptotic cells.

## RNA sequencing

At 24 and 72 h post-fatigue, a subset of paired hindlimbs were processed for RNA sequencing ($n = 6$ per sex per timepoint). Paired hindlimbs were immediately removed following expiration, trimmed of excess muscle and sub-merged in RNALater (Thermo Fisher Scientific, Waltham, MA, USA) at room temperature for 7 days to halt RNase activity and facilitate ACL removal without significant RNA degradation. Following this, paired knees were microdissected using stereomicroscopy (S Apo; Leica, Wetzlar, Germany) to remove each ACL. Extracted tissue was then individually flash frozen in liquid nitrogen, and stored at −80°C. To acquire RNA, individual ACLs were disrupted using a high-speed mechanical homogenizer (Model 150; Thermo Fisher Scientific) in lysis buffer (RLT; Qiagen, Hilden, Germany) and $\beta$-mercaptoethanol (Sigma-Aldrich, St Louis, MO, USA). Each specimen was then centrifuged, and the resulting supernatant was then processed for DNA digestion, RNA extraction and purification using a RNeasy Plus Micro Kit (Qiagen). Total RNA concentration and RNA integrity was quantified using a bioanalyzer (2100; Agilent Technologies, Santa Clara, CA, USA). All samples achieved an RNA integrity number of 7 or greater, and the mean RNA concentration was 1200 pg μL$^{-1}$. Because of the low concentration, two fatigued or contralateral control samples were pooled for all six samples (three left and three right ACL pools) acquired for each sex and post-injury timepoint, resulting in a total of 24 pools. Pools were then provided to the Indiana University School of Medicine Center for Medical Genomics for cDNA synthesis and library prep using an ultra-low input RNA kit (Smart-Seq v4; Takara Bio, Shiga,

Japan) followed by sequencing via a NovaSeq X system (Illumina, San Diego, CA, USA). After sequencing the quality of raw read data for each sample was checked using FastQC. (Andrew, 2010) Next, raw reads were aligned to the B6 mouse genome using STAR (Dobin et al., 2012) and then checked using FastQC. Gene expression quantification was then performed using featureCounts (Liao et al., 2014) to acquire counts of aligned reads for each gene, which was then used for differential gene expression analyses using edgeR (Huang et al., 2009). To correct for false discovery rate (FDR, $q$ values), $P$ values were adjusted for multiple hypotheses testing using the Benjamini–Hochberg procedure. Significant differentially expressed genes (DEGs) were defined as having a $q < 0.05$ and a log2 fold change greater than 1, relative to contra-lateral controls. Significant DEGs were uploaded to the Gene Ontology (http://geneontology.org) knowledgebase for taxonomic classification of biological processes, molecular functions and cellular components that involve sets of genes present in the dataset (Ashburner et al., 2000; The Gene Ontology Consortium, 2023; Thomas et al., 2022). Additionally, all read counts per million for each gene and sample were uploaded into ReactomeGSA (Griss et al., 2020) to identify significant differentially expressed pathways and functional groupings of genes both shared and unique to female and male datasets at each timepoint, as well as between timepoints for each sex. Pathway analysis was conducted without interactors using the Pathway Analysis with Down-Weighting of Overlapping Genes (PADOG) method, which gives more weight to genes that are gene-set specific rather to genes found in multiple gene sets (Tarca et al., 2012). Analysis did not include interactors or disease pathways. Significant differentially expressed pathways were defined as having a $q < 0.05$, a log2 fold change greater than 1 and at least four mapped genes.

## Statistical analysis

All statistical analyses were performed using Minitab, version 20 (Minitab LLC, State College, PA, USA) and Prism, version 10 (GraphPad Software Inc., San Diego, CA, USA). ClustVis was utilized for principal component analysis of all DEGs across post-injury timepoints and sex (Metsalu & Vilo, 2015). Normality of data was assessed using the Shapiro–Wilk test. To test for mechanical (knee hysteresis and stiffness) and histochemistry (denaturation and programmed cell death) differences between males and females at each post-injury timepoint, two-tailed, paired $t$ tests were used. All ACL and knee mechanical measures were adjusted for body weight using linear regression. Adjusted data was then compared between sexes and timepoints via ANOVA, after adjusting for body weight via linear regression. Data are reported as the

mean ± SD. An alpha of 0.05 was considered statistically significant.

## Results

### ACL and knee mechanical properties are sex-specific

From the load to ACL failure tests used to delineate the moderate (30% maximum force) and strenuous (60% maximum force) loads, the resulting data confirmed a significant sex difference in *in vivo* ACL strength ($P = 0.0477$). Female ACLs had a mean failure load of 12.67 ± 1.36 N, whereas the mean failure load was 15.70 ± 2.87 N for male ACLs (Fig. 2). Additionally, from the loading curves generated throughout fatigue testing, there were significant sex differences in knee stiffness at both the moderate ($P = 0.0094$) and strenuous ($P < 0.0001$) loads (Fig. 3*A*). At both loads, female knees were less stiff compared to males. Mean female knee stiffness at the moderate and strenuous loads was 21.56 ± 2.28 and 32.76 ± 3.74 N mm$^{-1}$, respectively, while mean male knee stiffness at both loads was 25.45 ± 4.48 and 42.53 ± 5.64 N mm$^{-1}$, respectively. Furthermore, males demonstrated significantly greater knee hysteresis at both the moderate ($P < 0.0001$) and strenuous ($P < 0.0001$) loads compared to females (Fig. 3*B*). Moderate and strenuous loading in males demonstrated mean knee hysteresis of 1.14 ± 0.13 and 3.14 ± 0.39 N*mm, respectively, compared to a female knee hysteresis for each load of 0.70 ± 0.09 and 2.11 ± 0.18 N*mm, respectively. This hysteretic outcome is expected because the load to ACL failure, and thus the applied 30% and 60% maximum force loads, is higher in males compared to females.

### Significant sex differences in accrued denatured collagen but not apoptosis

Twenty-four hours post-injury both female and male ACLs showed a significant increase in the percentage area of denatured collagen post-injury. There was a significant increase ($P = 0.0288$) in CHP-detected denatured collagen in fatigued female ACLs (9.99 ± 2.83%) compared to their contralateral control ACLs (0.74 ± 0.07%). Similarly, males also showed a significant increase ($P = 0.0530$) in denatured collagen in fatigued ACLs (6.23 ± 2.14%) compared to their contralateral control ACLs (0.96 ± 0.20%) (Fig. 4). Moreover, when fatigued female and male ACLs were adjusted to account for denatured collagen present in their paired control ACLs, females showed significantly more denatured collagen than males 24 h after fatigue loading ($P = 0.0479$).

In terms of caspase-3 (*Casp3*), both female and male fatigued ACLs demonstrated a significant percentage increase in apoptosis post-injury. There was a significant percentage increase ($P = 0.01$) in *Casp3*-detected apoptosis in fatigued female ACLs (2.98 ± 0.69) compared to their contralateral ACLs (1.87 ± 0.48). Similarly, fatigued male ACLs showed a higher percentage of *Casp3*-detected apoptosis (2.25 ± 0.74) compared to their contralateral ACLs (1.29 ± 0.18), although this did not reach significance ($P = 0.07$) (Fig. 5). When fatigued female and male ACLs were adjusted by their contra-

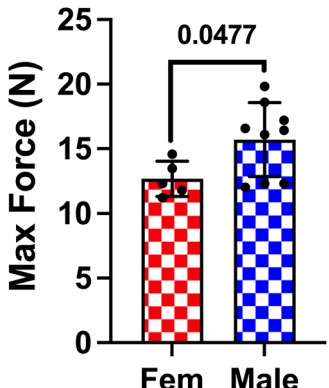

**Figure 2. Female mice have weaker ACLs than those of males**
Body weight adjusted mean and standard deviations for *in vivo* ACL strength. *n* = 10 mice per sex. Mean ± SD bodyweights are 20.15 ± 0.67 g for females and 25.85 ± 1.52 g for males.

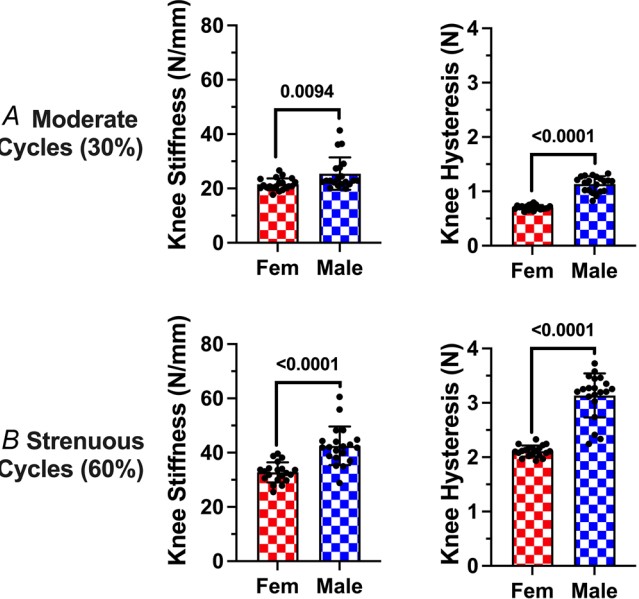

**Figure 3. Female mice have less knee stiffness and hysteresis compared to males**
Body weight adjusted mean and standard deviations for knee stiffness and hysteresis at (*A*) moderate and (*B*) strenuous loads. *n* = 20 mice per sex. Mean ± SD bodyweights are 19.40 ± 1.03 g for females and 26.35 ± 1.52 g for males.

lateral ACLs, there was no sex difference in the overall percentage of *Casp3* ($P = 0.86$).

### Males demonstrate a greater number of significant upregulated and downregulated DEGs compared to females

At 24 h post-injury, females had 13,405 mapped genes, of which 2032 of these met the cutoff criteria for significant DEGs relative to contralateral controls (1008 upregulated and 1024 downregulated) (Fig. 6*A* and *C*). By contrast, males had 14,739 mapped genes, with 2469 of these being differentially expressed relative to contralateral controls (1209 upregulated and 1260 downregulated) (Fig. 6*B* and *C*). Top 10 upregulated and downregulated female and male DEGs by FDR are listed in Table 1.

At 72 h post-injury, females had 12,911 mapped genes with 1053 (507 upregulated and 546 downregulated) genes significantly differentially expressed (Fig. 7*A* and *C*). By contrast, males had 14,739 mapped genes with 1297 (733 upregulated and 565 down regulated) of these being

significantly differentially expressed relative to contralateral controls. Top 10 upregulated and downregulated female and male DEGs by FDR are listed in Table 1. A list of all significant DEGs for males and females at 24 and 72 h post-injury are provided in the Supporting information (File S1).

Principal components analysis reinforced these gene expression differences between females and males. At 24 h post-injury, female control and fatigued significant DEGs demonstrate little separation, as opposed to the males (Fig. 8*A*). Principal component 1 (PC1) explains 43.6% of the variance and PC2 explains 22.9% of the variance across female and male datasets. With the addition of PC3 (19%), the total variance explained from the top 3 components is 86%. By contrast, at 72 h post-injury, both female and male fatigued datasets show clear separation from the control datasets (Fig. 8*B*). PC1 and PC2 explain 39% and 35.4% of the variance across female and male datasets, with the addition of PC3 explaining a total of 92% of the variance. All principal components and percentage variance explained at 24 and 72 h post-injury are provided in Tables 2 and 3, respectively.

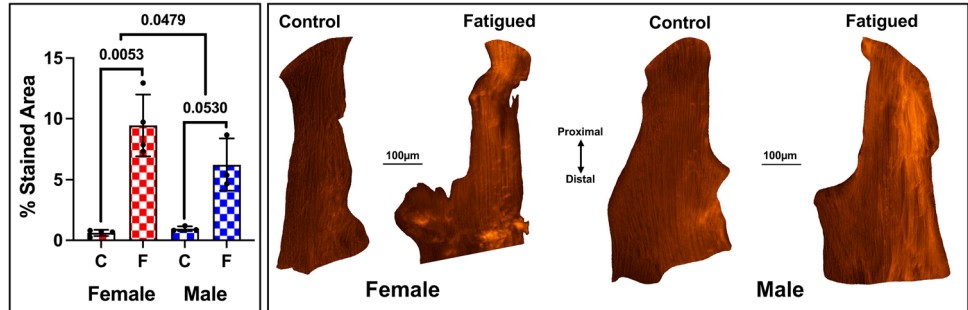

**Figure 4. Female ACLs accrue more fatigue-induced denatured collagen compared to males**
Left: mean percentage area of CHP-detected denatured collagen at 24 h post-injury. C = control ACL, F = fatigued ACL. *n* = 3–4 mice per sex. Each data point is the mean percentage across three to five tissue sections analyzed per knee. Right: paired male and female ACLs cropped from two mouse knees that represent the mean percentage area stained positive for denatured collagen.

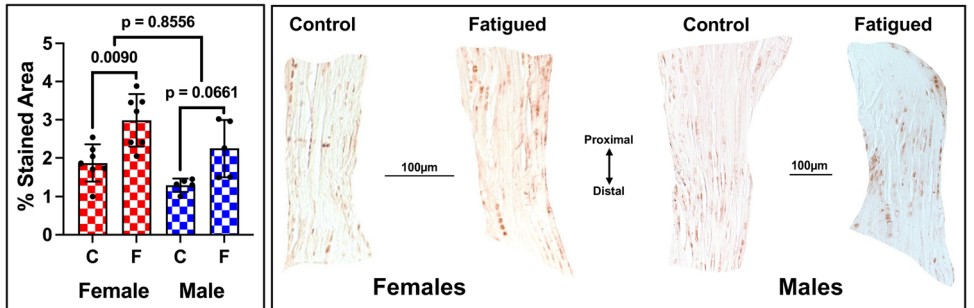

**Figure 5. Fatigued female ACLs exhibit significant cell apoptosis compared to their non-fatigued contra-lateral ACL**
Left: mean percentage area of Casp3-detected apoptosis post-injury. 24 h and 72 h are combined. C = control ACL, F = fatigued ACL. *n* = 6–7 mice per sex; Each data point is the mean percentage across four to six tissue sections per knee. Right: paired male and female ACLs cropped from two mouse knees that represent the mean percentage area stained positive for Casp3.

**Table 1. Top 10 upregulated and downregulated DEGs in ACLs 24- and 72-h post-injury.**

**24 h DEGs**

| | Upregulated | | | Downregulated | | |
|---|---|---|---|---|---|---|
| | Gene | Log2FC | FDR | Gene | Log2FC | FDR |
| **24 h Female DEGs** | | | | | | |
| | Col10a1 | 5.55 | $8.94 \times 10^{-45}$ | Lepr | −3.83 | $7.08 \times 10^{-43}$ |
| | Cthrc1 | 4.63 | $9.28 \times 10^{-41}$ | Abi3bp | −3.08 | $6.83 \times 10^{-38}$ |
| | Col2a1 | 2.99 | $5.24 \times 10^{-33}$ | Angptl7 | −3.73 | $2.54 \times 10^{-36}$ |
| | Nt5dc2 | 4.19 | $4.10 \times 10^{-24}$ | Cilp | −3.13 | $5.83 \times 10^{-35}$ |
| | Timp1 | 2.69 | $5.36 \times 10^{-16}$ | Clu | −3.20 | $8.64 \times 10^{-35}$ |
| | Col6a3 | 1.88 | $7.89 \times 10^{-15}$ | Igfbp5 | −2.70 | $1.61 \times 10^{-29}$ |
| | Lyz2 | 1.86 | $7.83 \times 10^{-14}$ | Chad | −2.86 | $1.06 \times 10^{-25}$ |
| | Il1rl1 | 4.39 | $2.71 \times 10^{-11}$ | Cilp2 | −2.71 | $4.62 \times 10^{-25}$ |
| | Postn | 3.20 | $8.95 \times 10^{-11}$ | Fbln1 | −3.76 | $7.18 \times 10^{-25}$ |
| | Stmn1 | 3.38 | $1.55 \times 10^{-10}$ | Cytl1 | −3.02 | $1.07 \times 10^{-24}$ |
| **24 h Male DEGs** | | | | | | |
| | Nt5dc2 | 4.18 | $5.72 \times 10^{-32}$ | Serping1 | −3.05 | $5.17 \times 10^{-19}$ |
| | Cks2 | 4.10 | $4.53 \times 10^{-21}$ | Clec3b | −4.70 | $3.36 \times 10^{-18}$ |
| | Uox | 12.54 | $5.64 \times 10^{-20}$ | Gstm1 | −2.85 | $2.93 \times 10^{-12}$ |
| | Cxcl3 | 11.05 | $5.17 \times 10^{-19}$ | Gm7694 | −10.61 | $3.24 \times 10^{-11}$ |
| | Cd300lf | 11.12 | $1.24 \times 10^{-15}$ | Mettl7a1 | −2.74 | $2.58 \times 10^{-10}$ |
| | Clec4d | 4.59 | $2.44 \times 10^{-15}$ | Nrep | −3.18 | $9.91 \times 10^{-10}$ |
| | Tcf19 | 3.82 | $6.28 \times 10^{-15}$ | Klf9 | −2.07 | $2.90 \times 10^{-9}$ |
| | Cthrc1 | 4.57 | $1.27 \times 10^{14}$ | Gabarapl1 | −1.94 | $7.35 \times 10^{-9}$ |
| | Rrm2 | 5.51 | $1.65 \times 10^{-14}$ | Rnf167 | −1.94 | $9.26 \times 10^{-9}$ |
| | Smc4 | 2.23 | $3.74 \times 10^{-14}$ | Ephx1 | −2.53 | $1.13 \times 10^{-8}$ |

**72 h DEGs**

| | Upregulated | | | Downregulated | | |
|---|---|---|---|---|---|---|
| | Gene | Log2FC | FDR | Gene | Log2FC | FDR |
| **72 h Female DEGs** | | | | | | |
| | Timp1 | 3.14 | $1.13 \times 10^{-50}$ | Kera | −5.18 | $3.03 \times 10^{-56}$ |
| | Mmp3 | 3.26 | $1.56 \times 10^{-41}$ | Pamr1 | −4.32 | $2.04 \times 10^{-42}$ |
| | Il1b | 10.02 | $7.61 \times 10^{-38}$ | Sepp1 | −3.30 | $6.21 \times 10^{-42}$ |
| | Il1rl1 | 5.08 | $1.71 \times 10^{-34}$ | Clec3b | −4.23 | $1.12 \times 10^{-40}$ |
| | Top2a | 4.15 | $1.92 \times 10^{-30}$ | Ecm2 | −3.54 | $9.98 \times 10^{-38}$ |
| | Inhba | 2.89 | $1.54 \times 10^{-29}$ | Thbs1 | −3.22 | $1.39 \times 10^{-34}$ |
| | Igfbp3 | 3.91 | $6.66 \times 10^{-29}$ | Igfbp5 | −2.86 | $5.32 \times 10^{-34}$ |
| | Mcm5 | 3.04 | $6.23 \times 10^{-28}$ | Cilp2 | −3.67 | $1.84 \times 10^{-32}$ |
| | Prc1 | 4.12 | $5.71 \times 10^{-25}$ | Htra4 | −2.76 | $2.60 \times 10^{-28}$ |
| | Uhrf1 | 3.62 | $8.13 \times 10^{-24}$ | Scara3 | −2.48 | $3.67 \times 10^{-28}$ |
| **72 h Male DEGs** | | | | | | |
| | Cxcl3 | 14.65 | $7.31 \times 10^{-32}$ | Clec3b | −5.04 | $1.78 \times 10^{-20}$ |
| | Nt5dc2 | 3.72 | $4.25 \times 10^{-26}$ | Serping1 | −3.15 | $2.03 \times 10^{-20}$ |
| | Pdpn | 3.61 | $1.33 \times 10^{-24}$ | Ephx1 | −3.60 | $3.18 \times 10^{-16}$ |
| | Clec4d | 6.17 | $2.69 \times 10^{-24}$ | Mettl7a1 | −3.43 | $1.28 \times 10^{-15}$ |
| | Mmp3 | 4.98 | $1.53 \times 10^{-22}$ | Gstm1 | −3.22 | $1.60 \times 10^{-15}$ |
| | Cd300lf | 13.03 | $6.75 \times 10^{-21}$ | Gas6 | −3.03 | $1.34 \times 10^{-14}$ |
| | Cks2 | 3.69 | $7.23 \times 10^{-18}$ | Pamr1 | −6.19 | $1.83 \times 10^{-12}$ |
| | Ier3 | 2.98 | $5.59 \times 10^{-16}$ | Fxyd1 | −2.68 | $2.09 \times 10^{-12}$ |
| | Ms4a6d | 3.45 | $9.20 \times 10^{-16}$ | Nrep | −3.59 | $3.27 \times 10^{-12}$ |
| | Glrx | 2.41 | $9.28 \times 10^{-16}$ | Fam124a | −2.93 | $5.80 \times 10^{-12}$ |

## Ontology of gene function and location suggest sex differences in the response to injury

Gene Ontology analyses of the 24 and 72 h post-injury datasets for females and males identified several over-represented processes that were unique to one another. At 24 h, significant ($q < 0.05$) female DEGs were skewed towards leukocyte migration, DNA replication, cell adhesion, migration and the cytoskeleton that were unique to their male counterparts. By contrast, male DEGs were uniquely over-represented in catabolic processes and protein synthesis, in addition to strong cell adhesion processes (Table 4).

At 72 h, male DEGs continued to favor DNA replication and mitotic processes. Interestingly, there were no unique immune signalling processes significantly over-represented. On the other hand, female DEGs continued to favor cytokine-signalling, along with strong proteolysis regulation and muscle cell development. Both sexes were over-represented in actin assembly, cell adhesion and ECM functions, but female processes appear to be coupled with muscle development, whereas males couple it with cell cycle regulation (Table 5). Full Gene Ontology lists for both sexes and post-injury timepoints provided in the Supporting information (File S2).

## Biomolecular pathways predicted by Reactome differ by sex across post-injury timepoints

Pathway analysis of significant ($P < 0.05$) female DEGs 24 h post-injury yielded 55 enriched biomolecular processes that were predicted from at least 3 genes (Fig. 9A). Enriched pathways upregulated among females ($n = 42$) are involved in repairing DNA, modulating the immune response, growth factor signalling and mRNA metabolism. Downregulated female pathways ($n = 13$) are related to ECM/cytoskeleton anchorage, keratinization and mechanosignaling. In males, 110 significantly enriched pathways were identified 24 h post-injury (Fig. 9B). Enriched pathways upregulated among males ($n = 107$) are involved in immune response activation, regulating apoptosis, growth factor signalling and RNA processing. Downregulated male pathways ($n = 3$) are related to insulin processing and transcription.

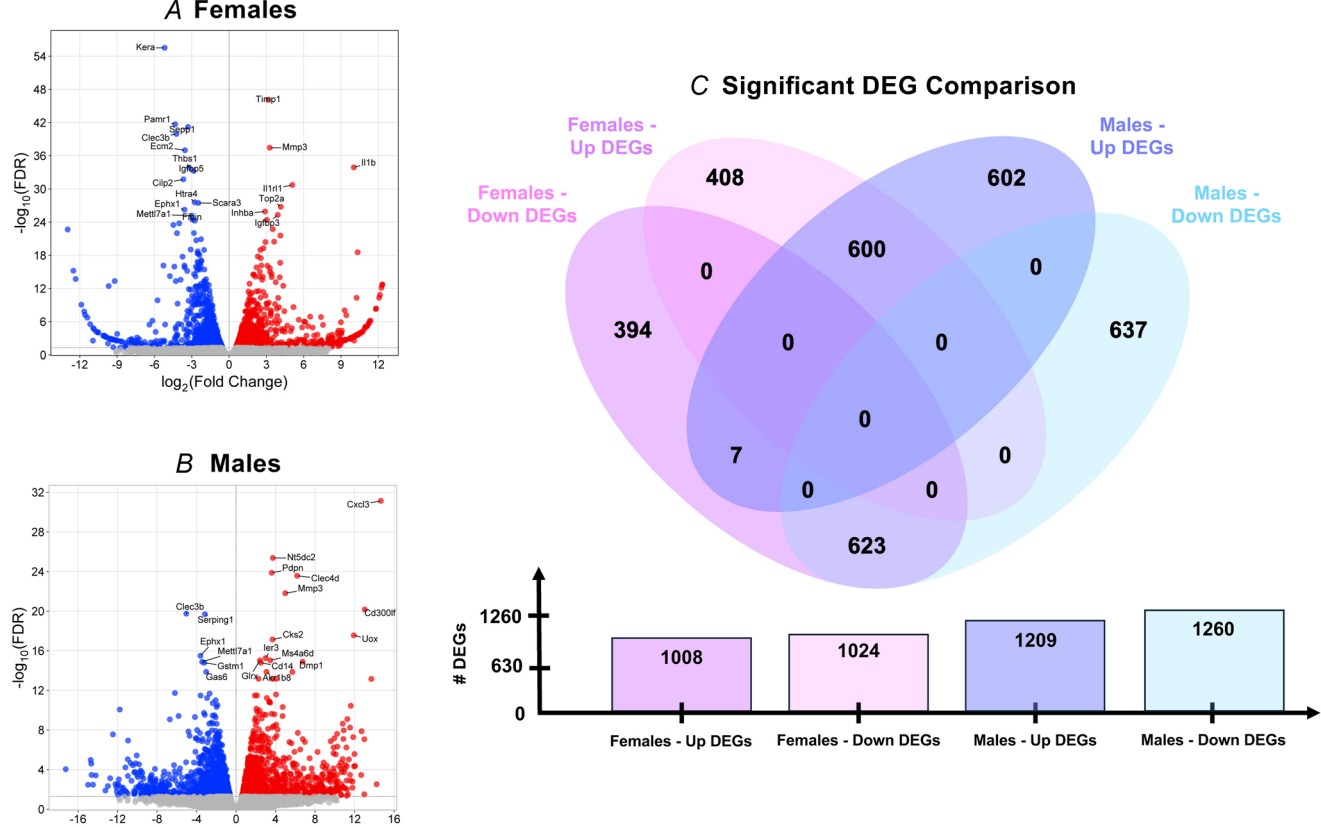

**Figure 6. 24-h DEG comparisons**
Volcano plots by FDR and log2 fold change for (*A*) females and (*B*) males. *C*, venn diagram comparing significant ($q < 0.05$) upregulated and downregulated DEGs between females and males.

At 72 h post-injury, 24 significantly enriched female pathways were identified (Fig. 9*C*). Enriched upregulated pathways among females ($n = 17$) are involved in the immune response, heat stress and ECM remodelling. Downregulated female pathways ($n = 7$) are related to metabolism and immune signalling. In males, 55 enriched

pathways were identified (Fig. 9*D*). Enriched pathways upregulated in males ($n = 43$) are related to cell cycle regulation, responding to DNA damage and *Rho* GTPase signalling.

Downregulated male pathways ($n = 12$) are involved in neural function and metabolism. All significantly

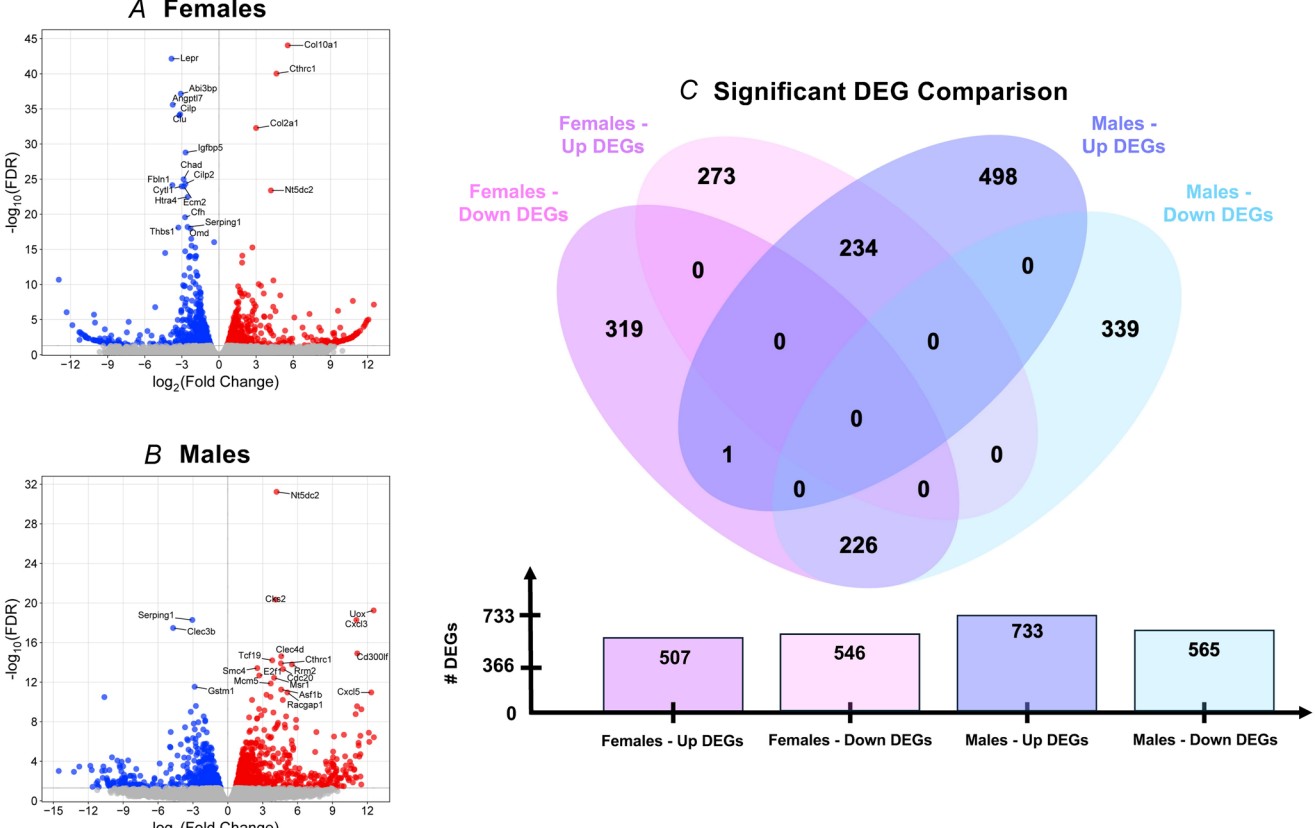

**Figure 7. 72-h DEG comparisons**
Volcano plots by FDR and log2 fold change for (*A*) females and (*B*) males. *C*, venn diagram comparing significant (*q* < 0.05) upregulated and downregulated DEGs between females and males.

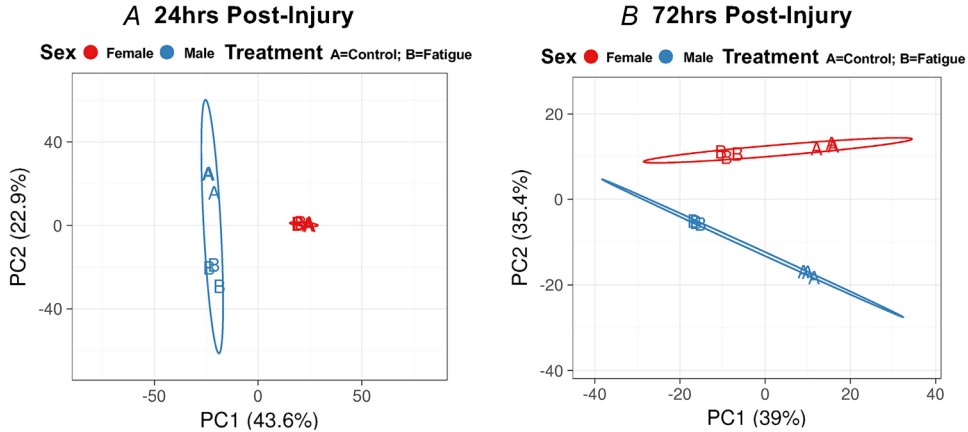

**Figure 8. Principal component analysis**
Plots of PC1 and PC2 of log2 CPMs for male and female DEGs at (*A*) 24 h and (*B*) 72 h post-injury.

**Table 2. Principal components and percentage variance explained by principal components for all significant DEGs shared by female and male ACLs 24 h post-injury.**

|  | PC1 | PC2 | PC3 | PC4 | PC5 | PC6 | PC7 | PC8 | PC9 |
|---|---|---|---|---|---|---|---|---|---|
| **24 h Principal components** | | | | | | | | | |
| **Female control 1** | −24.73 | 0.92 | −17.92 | −0.48 | −0.90 | −2.96 | 0.61 | 8.02 | −3.42 |
| **Female control 2** | −24.00 | 0.25 | −18.57 | −0.23 | −0.14 | −5.97 | −1.04 | −7.00 | −0.80 |
| **Female control 3** | −24.87 | 0.01 | −21.86 | −0.22 | −0.06 | 8.93 | 0.84 | −1.10 | 3.17 |
| **Female fatigue 1** | −20.50 | 1.13 | 19.04 | −0.61 | −0.32 | −4.07 | −0.52 | 2.20 | 7.61 |
| **Female fatigue 2** | −18.91 | 0.69 | 20.40 | −0.11 | 0.44 | −1.02 | −1.06 | −1.82 | −3.94 |
| **Female fatigue 3** | −19.35 | 0.79 | 25.91 | −0.28 | 0.35 | 5.03 | 1.48 | −0.55 | −2.78 |
| **Male control 1** | 24.20 | 24.56 | −2.06 | −7.01 | 5.08 | 0.55 | 0.20 | 0.01 | −1.53 |
| **Male control 2** | 23.62 | 24.80 | −1.68 | −6.92 | 4.97 | −0.41 | −0.99 | 0.13 | 1.56 |
| **Male control 3** | 21.16 | 15.52 | −0.63 | 20.90 | −8.22 | −0.13 | 1.26 | −0.15 | 0.08 |
| **Male fatigue 1** | 23.55 | −20.46 | −0.30 | −9.31 | −8.08 | −1.17 | 7.88 | −0.70 | 0.20 |
| **Male fatigue 2** | 18.57 | −29.23 | −1.71 | 8.64 | 15.07 | −0.30 | 0.31 | 0.35 | 0.14 |
| **Male fatigue 3** | 21.26 | −18.97 | −0.61 | −4.37 | −8.20 | 1.52 | −8.97 | 0.61 | −0.29 |
| **Explained variance** | | | | | | | | | |
| **Individual** | 0.44 | 0.23 | 0.19 | 0.05 | 0.04 | 0.01 | 0.01 | 0.01 | 0.01 |
| **Cumulative** | 0.44 | 0.66 | 0.86 | 0.91 | 0.95 | 0.96 | 0.97 | 0.98 | 0.99 |

**Table 3. Principal components and percentage variance explained by principal components for all significant DEGs shared by female and male ACLs 72 h post-injury.**

|  | PC1 | PC2 | PC3 | PC4 | PC5 | PC6 | PC7 | PC8 | PC9 |
|---|---|---|---|---|---|---|---|---|---|
| **72 h Principal components** | | | | | | | | | |
| **Female control 1** | −16.03 | 12.43 | 8.15 | −2.39 | −3.41 | 3.96 | −1.41 | 0.10 | −1.94 |
| **Female control 2** | −15.64 | 13.00 | 7.91 | −0.15 | 6.20 | 0.01 | 0.42 | 1.31 | 1.61 |
| **Female control 3** | −12.08 | 11.80 | 4.38 | 2.22 | −3.75 | −4.92 | 1.43 | −1.11 | 0.80 |
| **Female fatigue 1** | 10.38 | 11.04 | −13.32 | −6.73 | 0.11 | −1.18 | 0.20 | −0.78 | 0.84 |
| **Female fatigue 2** | 6.54 | 10.55 | −8.81 | 3.69 | 2.39 | 0.21 | −1.37 | −3.38 | −2.54 |
| **Female fatigue 3** | 9.10 | 9.80 | −12.07 | 3.74 | −1.66 | 2.02 | 0.83 | 3.88 | 1.09 |
| **Male control 1** | −10.03 | −17.04 | −4.81 | 0.54 | −0.89 | 1.16 | −2.45 | −1.80 | 3.81 |
| **Male control 2** | −9.05 | −16.91 | −4.60 | −0.27 | 0.68 | −2.67 | −2.96 | 2.50 | −2.49 |
| **Male control 3** | −11.53 | −18.31 | −5.38 | −0.42 | 0.45 | 1.37 | 4.91 | −0.64 | −1.30 |
| **Male fatigue 1** | 16.30 | −5.43 | 9.50 | −1.41 | −0.06 | −1.67 | 0.40 | 1.45 | −0.73 |
| **Male fatigue 2** | 15.19 | −5.83 | 9.25 | 0.58 | 0.17 | 1.16 | 0.55 | −1.64 | −0.02 |
| **Male fatigue 3** | 16.86 | −5.10 | 9.81 | 0.60 | −0.22 | 0.56 | −0.54 | 0.11 | 0.87 |
| **Explained variance** | | | | | | | | | |
| **Individual** | 0.39 | 0.35 | 0.18 | 0.02 | 0.01 | 0.01 | 0.01 | 0.01 | 0.01 |
| **Cumulative** | 0.39 | 0.74 | 0.92 | 0.94 | 0.95 | 0.96 | 0.97 | 0.98 | 0.99 |

enriched female and male pathways containing at least 3 DEGs at 24- and 72-h post-injury are provided in the Supporting information (File S3).

## Discussion

Similar to humans, adolescent B6 female mice demonstrated up to 23% lower anterior knee stiffness (i.e. greater laxity) under load compared to males (Shultz et al., 2008). This outcome also mirrors reporting by others showing an *ex vivo* female bias towards greater anterior tibial displacement in young adult B6 mice (Liu et al., 2023). Females also had 24% weaker ACLs, compared to males, meaning less resistance to anterior tibial translation under multi-axial forces (Butler et al., 1980). Part of our hypothesis was that this increased female knee laxity would result in greater fatigue-induced ACL matrix damage, relative to males. This hypothesis was confirmed. Collagen denaturation among fatigued female ACLs was 43% more than that among fatigued male ACLs and was largely concentrated within the anterior proximal third of the ACL, as we have reported

**Table 4. Unique Gene Ontologies (GO) of significant DEGs for females and males 24 h post-injury.**

| GO | Terms | # Ref Genes | # Obs/Exp Genes | Fold | FDR |
|---|---|---|---|---|---|
| **Unique 24 h female GOs** | | | | | |
| **BP** 0000727 | Double-strand break repair | 11 | 7/1 | 7.01 | $2.11 \times 10^{-2}$ |
| 0006271 | DNA strand elongation | 11 | 7/1 | 7.01 | $2.11 \times 10^{-2}$ |
| 0035335 | Peptidyl-tyrosine dephosphorylation | 19 | 9/1.7 | 5.22 | $2.88 \times 10^{-2}$ |
| 0050900 | Leukocyte migration | 34 | 14/3.1 | 4.54 | $1.05 \times 10^{-3}$ |
| 0044772 | Mitotic cell cycle phase transition | 37 | 14/3.4 | 4.17 | $3.55 \times 10^{-3}$ |
| 051783 | Regulation of nuclear division | 33 | 12/3 | 4.01 | $3.08 \times 10^{-2}$ |
| 0071900 | Regulation of serine/threonine kinase activity | 54 | 16/5 | 3.27 | $2.63 \times 10^{-2}$ |
| 0006954 | Inflammatory response | 93 | 27/8.4 | 3.20 | $5.56 \times 10^{-5}$ |
| 0048285 | Organelle fission | 107 | 24/9.7 | 2.47 | $4.48 \times 10^{-2}$ |
| 2000026 | Regulation of multicellular organismal development | 142 | 31/12.9 | 2.41 | $8.11 \times 10^{-3}$ |
| 0051276 | Chromosome organization | 138 | 30/12.5 | 2.40 | $1.25 \times 10^{-2}$ |
| 0050793 | Regulation of developmental process | 251 | 46/22.8 | 2.02 | $6.89 \times 10^{-3}$ |
| 0051239 | Regulation of multicellular organismal process | 328 | 58/29.8 | 1.95 | $1.47 \times 10^{-3}$ |
| 0048870 | Cell motility | 301 | 53/27.3 | 1.94 | $4.34 \times 10^{-3}$ |
| 0051128 | Regulation of cellular component organization | 372 | 59/33.7 | 1.75 | $4.10 \times 10^{-2}$ |
| 1901135 | Carbohydrate derivative metabolic process | 457 | 69/41.5 | 1.66 | $4.22 \times 10^{-2}$ |
| 0006950 | Response to stress | 876 | 118/79.5 | 1.48 | $2.47 \times 10^{-2}$ |
| 1901564 | Organonitrogen compound metabolism | 2304 | 294/209 | 1.41 | $8.10 \times 10^{-7}$ |
| 0019538 | Protein metabolic process | 1803 | 215/163.6 | 1.31 | $3.85 \times 10^{-2}$ |
| **MF** 0017116 | Single-stranded DNA helicase activity | 10 | 6/0.9 | 6.61 | $4.70 \times 10^{-2}$ |
| 0008094 | ATP-dependent activity, acting on DNA | 50 | 15/4.5 | 3.31 | $1.29 \times 10^{-2}$ |
| 0005125 | Cytokine activity | 96 | 22/8.7 | 2.53 | $2.13 \times 10^{-2}$ |
| 0022836 | * Gated channel activity * | 238 | 5/21.6 | 0.23 | $1.10 \times 10^{-2}$ |
| **CC** 0005657 | Replication fork | 23 | 9/2.1 | 4.31 | $4.44 \times 10^{-2}$ |
| 0005874 | Microtubule | 148 | 29/13.4 | 2.16 | $2.97 \times 10^{-2}$ |
| 0099513 | Polymeric cytoskeletal fibre | 242 | 45/21.9 | 2.05 | $1.82 \times 10^{-3}$ |
| 0005929 | * Cilium * | 256 | 7/23.2 | 0.3 | $4.39 \times 10^{-2}$ |
| 0034702 | * Monoatomic ion channel complex * | 174 | 2/15.8 | 0.13 | $1.08 \times 10^{-2}$ |
| 1902495 | * Transmembrane transporter complex * | 220 | 2/20 | 0.1 | $1.58 \times 10^{-4}$ |
| **Unique 24 h male Gos** | | | | | |
| **BP** 0010811 | Positive regulation of cell-substrate adhesion | 9 | 7/1 | 7.22 | $1.16 \times 10^{-2}$ |
| 0006270 | DNA replication initiation | 17 | 9/1.6 | 4.91 | $2.72 \times 10^{-4}$ |
| 0042273 | Ribosomal large subunit biogenesis | 47 | 18/5.1 | 3.55 | $3.53 \times 10^{-2}$ |
| 0007160 | Cell-matrix adhesion | 43 | 15/4.6 | 3.24 | $2.22 \times 10^{-2}$ |
| 0072329 | Monocarboxylic acid catabolic process | 47 | 16/5.1 | 3.16 | $1.69 \times 10^{-2}$ |
| 0051241 | Negative regulation of multicellular organismal process | 53 | 17/5.7 | 2.98 | $7.68 \times 10^{-11}$ |
| 0044242 | Cellular lipid catabolic process | 89 | 25/9.6 | 2.61 | $2.93 \times 10^{-2}$ |
| 0046395 | Carboxylic acid catabolic process | 94 | 26/10.1 | 2.57 | $1.83 \times 10^{-3}$ |
| 0044282 | Small molecule catabolic process | 120 | 31/12.9 | 2.40 | $2.98 \times 10^{-2}$ |
| 0007015 | Actin filament organization | 148 | 34/15.9 | 2.13 | $7.92 \times 10^{-3}$ |
| 0022613 | Ribonucleoprotein complex biogenesis | 213 | 48/22.9 | 2.09 | $3.15 \times 10^{-7}$ |
| 0071495 | cellular response to endogenous stimulus | 212 | 46/22.8 | 2.01 | $2.75 \times 10^{-5}$ |
| 0051336 | Regulation of hydrolase activity | 240 | 51/25.9 | 1.97 | $1.88 \times 10^{-5}$ |
| 0044248 | Cellular catabolic process | 428 | 86/46.1 | 1.86 | $1.43 \times 10^{-4}$ |
| 0035556 | Intracellular signal transduction | 463 | 82/49.9 | 1.64 | $0.00 \times 10^{+0}$ |
| 0065009 | Regulation of molecular function | 515 | 87/55.5 | 1.57 | $9.24 \times 10^{-4}$ |
| 0031325 | Positive regulation of cellular metabolic process | 581 | 97/62.6 | 1.55 | $1.60 \times 10^{-4}$ |
| 0002443 | * Leukocyte mediated immunity * | 196 | 2/21.1 | 0.09 | $8.12 \times 10^{-21}$ |

*(Continued)*

**Table 4. (Continued)**

|  | GO | Terms | # Ref Genes | # Obs/Exp Genes | Fold | FDR |
|---|---|---|---|---|---|---|
| **MF** | 0003735 | Structural constituent of ribosome | 108 | 26/11.6 | 2.23 | $3.72 \times 10^{-2}$ |
|  | 0097367 | Carbohydrate derivative binding | 275 | 58/29.6 | 1.96 | $3.44 \times 10^{-4}$ |
|  | 0140678 | Molecular function inhibitor activity | 181 | 38/19.5 | 1.95 | $2.96 \times 10^{-2}$ |
|  | 1901265 | Nucleoside phosphate binding | 262 | 53/28.2 | 1.88 | $4.14 \times 10^{-3}$ |
|  | 1901363 | Heterocyclic compound binding | 287 | 57/30.9 | 1.84 | $2.77 \times 10^{-3}$ |
|  | 0043168 | Anion binding | 353 | 67/38 | 1.76 | $2.04 \times 10^{-3}$ |
|  | 0016787 | Hydrolase activity | 1436 | 205/154.7 | 1.32 | $1.06 \times 10^{-2}$ |
| **CC** | 0005832 | Chaperonin-containing T-complex | 8 | 6/0.9 | 6.96 | $1.55 \times 10^{-2}$ |
|  | 0000940 | Outer kinetochore | 9 | 6/1 | 6.19 | $4.23 \times 10^{-2}$ |
|  | 0022626 | Cytosolic ribosome | 79 | 28/8.5 | 3.29 | $2.29 \times 10^{-6}$ |
|  | 1990904 | Ribonucleoprotein complex | 418 | 73/45 | 1.62 | $1.32 \times 10^{-2}$ |

Column 1 (C1) taxonomic classes are BP = biological process, MF = molecular function, CC = cellular component. C2 comprises gene ontology identifiers. C3 comprises terms. C4 comprises total number of reference genes in term. C5 comprises the number of genes observed in the dataset and number expected. C6 is the fold enrichment score and C7 comprises the false discovery rate calculated via Benjamin–Hochberg multiple comparisons test.

previously (Loflin et al., 2023). Both females and males showed increased apoptotic activity in response to ACL overuse; however, a significant sex difference was not observed.

Greater accrual of fatigue-induced collagen matrix damage among female ACLs was further reflected in their downregulation of extracellular matrix genes (*Kera*, *Cilp2*, *Thbs1* and *Ecm2*). We also hypothesized a delayed or prolonged, physiological response to these collagen disruptions, relative to male ACLs. Although the outcomes of this study cannot definitively confirm or reject this part of the hypothesis, there is evidence to support a sex-based difference in the initial physiological response to ACL overuse. At 24 h post-injury, enriched pathways from female ACLs emphasize DNA repair and immune modulation. Featured DNA repair mechanisms and top differentially expressed genes (*Top2a*, *Mcm5*, *Prc1* and *Uhrf1*) indicate homologous recombination and the maintenance and/or restoration of genomic integrity during cell division. In response, females are also focused on regulating cell proliferation via clathrin-mediated endocytosis. This pathway modulates proliferative activity by recycling endothelial growth factor receptors (*Egfr*) to prevent overaction of cell proliferative down-stream signals. Concurrently, immune modulation is manifested through the signalling of *Il-10*, a potent anti-inflammatory cytokine, and is probably antagonizing the strong upregulation of *Il-1b* in conjunction with an anti-inflammatory interleukin receptor, *Il-1rl1* (Dinarello, 2011). *Il-10* can also assist in the regulation of *Dap12*, an adaptor protein, and rhodopsin-like receptors involved in signalling immune cell activation for the clearance of cellular debris and pathogens in damaged tissue regions (Sun & Ye, 2012; Turnbull & Colonna, 2007). Further

regulation of both cell proliferative activities and the inflammatory response is probably being provided via oestrogen receptor signalling. Oestrogen signalling can assist in reducing inflammation via inhibition of nuclear factor-$\kappa$B signalling (Liu et al., 2017; Stein & Yang, 1995), although potentially at the detriment of cell proliferation because there is some evidence suggesting a progressive decrease in ACL fibroblast proliferation with oestrogen exposure (Liu et al., 1997; Yu et al., 1999).

Male pathways at 24 h post-injury indicate a dynamic environment consisting of immune activation, cytoskeletal organization, cell cycle regulation and hormonal signalling. Similar to females, males show significant upregulation of *Il-10* to modulate their strong immune response, in part driven by *Cxcl3*, *Clec4d*, and *Cd300lf* pro-inflammatory genes. Upregulated pathways involved in the male immune response include neutrophil degranulation, antigen recognition and interferon signalling, which are all involved in the removal of cellular debris and cellular stress defense mechanisms (Ethuin et al., 2004). These processes are co-ordinated by cytokines and probably facilitated by cytoskeletal reorganization pathways involved in cell migration and phagocytosis. This is evident with the upregulation of actin dynamics, Fc gamma receptor (*Fcgr*) driven phagocytosis and *Rho* GTPases to enable immune cells to engulf and clear damaged cells and pathogens. To enhance and regulate the immune response, *Jak-Stat* and *Mapk* signalling cascades are enriched. Further support for immune cell proliferation and survival is provided by the upregulation of the *Gab1* signalosome and *Egfr* signalling, which, as for females, appears to be facilitated by clathrin-mediated endocytosis. Males also show significant upregulation of oestrogen-dependent signalling potentially to modulate

**Table 5. Unique Gene Ontologies (GO) of significant DEGs for females and males 72 h post-injury.**

| | GO | Terms | # Ref Genes | # Obs/Exp Genes | Fold | FDR |
|---|---|---|---|---|---|---|
| **Unique 72 h female Gos** | | | | | | |
| **BP** | 0030239 | Myofibril assembly | 28 | 14/1.3 | 10.58 | $9.72 \times 10^{-9}$ |
| | 0010927 | Cellular component assembly | 33 | 14/1.6 | 8.98 | $1.59 \times 10^{-7}$ |
| | 0051146 | Striated muscle cell differentiation | 39 | 15/1.8 | 8.14 | $1.82 \times 10^{-7}$ |
| | 0031032 | Actomyosin structure organization | 60 | 15/2.8 | 5.29 | $1.51 \times 10^{-4}$ |
| | 0010951 | Negative regulation of endopeptidase activity | 62 | 14/2.9 | 4.78 | $1.56 \times 10^{-3}$ |
| | 0010466 | Negative regulation of peptidase activity | 64 | 14/3 | 4.63 | $2.36 \times 10^{-3}$ |
| | 0045861 | Negative regulation of proteolysis | 69 | 15/3.3 | 4.60 | $1.08 \times 10^{-3}$ |
| | 0051346 | Negative regulation of hydrolase activity | 69 | 14/3.3 | 4.29 | $6.11 \times 10^{-3}$ |
| | 0048646 | Anatomical structure formation | 109 | 21/5.1 | 4.08 | $6.12 \times 10^{-5}$ |
| | 0140694 | Organelle assembly | 121 | 23/5.7 | 4.02 | $1.80 \times 10^{-5}$ |
| | 0019221 | Cytokine-mediated signalling | 100 | 17/4.7 | 3.60 | $7.44 \times 10^{-3}$ |
| | 0034097 | Response to cytokine | 149 | 24/7 | 3.41 | $2.41 \times 10^{-4}$ |
| | 0007015 | Actin filament organization | 148 | 21/7 | 2.98 | $4.26 \times 10^{-4}$ |
| | 0032989 | Cellular anatomical entity | 199 | 28/9.4 | 2.60 | $2.89 \times 10^{-4}$ |
| | 0097435 | Supramolecular fibre organization | 293 | 36/13.9 | 0.42 | $3.27 \times 10^{-2}$ |
| | 0090304 | * Nucleic acid metabolic process * | 902 | 18/42.6 | 0.12 | $2.88 \times 10^{-2}$ |
| | 0051606 | * Detection of stimulus * | 346 | 2/16.3 | 0.12 | $4.32 \times 10^{-5}$ |
| | 0007600 | * Sensory perception * | 538 | 3/25.4 | 0.12 | $4.32 \times 10^{-5}$ |
| **MF** | 0005201 | ECM constituent | 54 | 15/2.5 | 5.88 | $1.01 \times 10^{-5}$ |
| | 0019838 | Growth factor binding | 44 | 10/2.1 | 4.81 | $1.70 \times 10^{-2}$ |
| | 0003735 | Structural constituent of ribosome | 108 | 20/5.1 | 3.92 | $8.07 \times 10^{-5}$ |
| | 0043167 | * Ion binding * | 619 | 52/29.2 | 1.78 | $3.77 \times 10^{-2}$ |
| **CC** | 0022627 | Cytosolic small ribosomal subunit | 30 | 12/1.4 | 8.46 | $1.98 \times 10^{-6}$ |
| | 0043292 | Contractile fibre | 76 | 22/3.6 | 6.12 | $1.52 \times 10^{-9}$ |
| **Unique 72 h Male Gos** | | | | | | |
| **BP** | 0007076 | Mitotic chromosome condensation | 6 | 5/0.3 | 14.57 | $6.15 \times 10^{-3}$ |
| | 0000727 | Double-strand break repair | 11 | 6/0.6 | 9.54 | $2.21 \times 10^{-2}$ |
| | 0006270 | DNA replication initiation | 17 | 7/1 | 7.20 | $4.09 \times 10^{-2}$ |
| | 0000070 | Mitotic sister chromatid segregation | 41 | 15/2.3 | 6.40 | $5.80 \times 10^{-6}$ |
| | 0000079 | Regulation of serine/threonine kinase activity | 25 | 9/1.4 | 6.29 | $1.00 \times 10^{-2}$ |
| | 0045785 | Positive regulation of cell adhesion | 28 | 10/1.6 | 6.24 | $3.26 \times 10^{-3}$ |
| | 0098813 | Nuclear chromosome segregation | 62 | 18/3.5 | 5.08 | $1.13 \times 10^{-5}$ |
| | 0007052 | Mitotic spindle organization | 35 | 10/2 | 5.00 | $3.16 \times 10^{-2}$ |
| | 0006261 | DNA-templated DNA replication | 61 | 16/3.5 | 4.59 | $3.72 \times 10^{-4}$ |
| | 1901990 | Regulation of mitotic cell cycle phase transition | 63 | 16/3.6 | 4.44 | $6.03 \times 10^{-4}$ |
| | 0000280 | Nuclear division | 93 | 21/5.3 | 3.95 | $8.53 \times 10^{-5}$ |
| | 0048285 | Organelle fission | 107 | 22/6.1 | 3.59 | $2.45 \times 10^{-4}$ |
| | 0001932 | Regulation of protein phosphorylation | 150 | 23/8.6 | 2.68 | $2.58 \times 10^{-2}$ |
| | 0016477 | Cell migration | 246 | 34/14.1 | 2.42 | $3.86 \times 10^{-3}$ |
| | 0048870 | Cell motility | 301 | 37/17.2 | 2.15 | $2.35 \times 10^{-2}$ |
| | 0051128 | Regulation of cellular component organization | 372 | 43/21.3 | 2.02 | $2.28 \times 10^{-2}$ |
| | 0003008 | * System process * | 658 | 14/37.6 | 0.37 | $1.41 \times 10^{-2}$ |
| **MF** | 0005200 | Structural constituent of cytoskeleton | 36 | 10/2.1 | 4.86 | $1.31 \times 10^{-2}$ |
| | 0019887 | Protein kinase regulator activity | 89 | 17/5.1 | 3.34 | $5.67 \times 10^{-3}$ |
| | 0097367 | Carbohydrate derivative binding | 275 | 38/15.7 | 2.42 | $2.87 \times 10^{-4}$ |
| | 0030234 | Enzyme regulator activity | 556 | 59/31.8 | 1.86 | $2.55 \times 10^{-3}$ |
| | 0044877 | Protein-containing complex binding | 582 | 60/33.3 | 1.8 | $6.17 \times 10^{-3}$ |
| | 0003823 | * Antigen binding * | 227 | 1/13 | 0.08 | $2.14 \times 10^{-2}$ |

*(Continued)*

**Table 5. (Continued)**

| | GO | Terms | # Ref Genes | # Obs/Exp Genes | Fold | FDR |
|---|---|---|---|---|---|---|
| **CC** | 0000940 | Outer kinetochore | 9 | 5/0.5 | 9.71 | $2.72 \times 10^{-2}$ |
| | 0005657 | Replication fork | 23 | 8/1.3 | 6.08 | $1.10 \times 10^{-2}$ |
| | 0000779 | Condensed chromosome, centromeric | 47 | 15/2.7 | 5.58 | $1.21 \times 10^{-5}$ |
| | 1902554 | Serine/threonine kinase complex | 65 | 14/3.7 | 3.77 | $6.30 \times 10^{-3}$ |
| | 0005819 | Spindle | 76 | 16/4.3 | 3.68 | $2.15 \times 10^{-3}$ |
| | 1902911 | Protein kinase complex | 74 | 15/4.2 | 3.54 | $6.85 \times 10^{-3}$ |
| | 0015630 | Microtubule cytoskeleton | 446 | 49/25.5 | 1.92 | $5.04 \times 10^{-3}$ |

Column 1 (C1) taxonomic classes are BP = biological process, MF = molecular function, CC = cellular component. C2 comprises gene ontology identifiers. C3 comprises terms. C4 comprises total number of reference genes in term. C5 is the number of genes observed in the dataset and number expected. C6 comprises the fold enrichment score and C7 is the false discovery rate calculated via Benjamin–Hochberg multiple comparisons test.

the inflammatory response in a similar fashion to that of females. However, males have significantly less circulating oestrogen and immune cell oestrogen receptors compared to females, probably resulting in a more limited effect on resolving the inflammatory response (Hutson et al., 2019).

At 72 h post-injury, females continue to prioritize modulation of the immune response by balancing anti-inflammatory *Il-10* and *Il-1rl1* signalling and pro-inflammatory *Clec7a* and *Lyz2* signalling. This uptick in inflammation is probably a direct result of regulated necrosis, which involves the lysing of damaged cells and the release of damage-associated molecular patterns (i.e. DAMPs) (Murao et al., 2021). Upregulated pathways also indicate continued cellular stress with the activation of the heat shock response by *Hsf1*, and the arrest of cell cycle progression via the inhibition of proteolytic activity by the anaphase-promoting complex (APC/C), thus preventing premature mitotic progression. Additional pathways suggestive of further immune modulation and cellular communication revolve around the upregulation of G-coupled protein receptors (*Gpcrs*), such as rhodopsin-like receptors, and ligand binding. There also may be an attempt to repair damage fibrocartilage within the proximal ACL enthesis, where the majority of denatured collagen was observed, with the upregulation of key cartilage matrix genes *Col10a1* and *Col2a1*. However, *Cytl1*, a regulator of chondrogenic differentiation of mesenchymal cells (Kim et al., 2007), is downregulated and there were no upregulated ECM remodelling pathways that were enriched.

By contrast to females, males 72 h post-injury prioritize mitotic activities and rapid cell proliferation with the significant upregulation of *Nt5dc2*, *Cks2*, *Rrm2*, *Tcf19* and *Smc4*, along with enriched pathways involved in mitotic spindle formation, chromosome segregation, nuclear envelope breakdown and reassembly, and cyclin A, B1 and B2 events during the G2/M transition. Together,

these processes indicate strong cell division activity and are further supported by APC/C:*Cdc20*-mediated degradation of cyclin B, triggering the exit from mitosis. Males are not only emphasizing cell proliferation, but also cell migration with the upregulation of *Rho* GTPase formins that nucleate actin filaments essential for keratinocyte, fibroblast and endothelial cell movements, and *Rho* GTPase effectors that co-ordinate these processes. However, the heightened cell proliferation and migration probably resulted in DNA replication stress because both ataxia-telangiectasia and Rad3-related kinase (*Atr*), an activator of the DNA damage response, and protein SUMOylation, a post-transcriptional modification that facilitates the repair of genomic lesions, are simultaneously upregulated. Moreover, there are also indicators of endoplasmic reticulum stress related to misfolded proteins with the activation of the unfolded protein response (Hetz, 2012). Collectively, replication and endoplasmic reticulum stress suggest robust cell proliferation and protein synthesis as cells adapt to stress conditions (e.g. hypoxia) at the repair sites.

Overall, an acute ACL overuse injury results in greater collagen matrix damage in females compared to males, even with the applied loads to ACL being proportional to the sex-specific difference in ACL strength. This difference is probably an effect of higher circulating oestrogen levels in females, which has been associated with an increase in knee laxity, a decrease in ACL stiffness and a reduction in collagen cross-links, and thus greater susceptibility for ACL injury (Lee et al., 2015; Shultz et al., 2005; Slauterbeck et al., 1999). Despite females experiencing greater tissue damage, the biological response to these collagen disruptions at 24 h post-injury demonstrates a controlled reparative process that prioritizes the restoration of DNA integrity at the same time as minimizing inflammatory activity. This is similar to what occurs in female innate immunity across species where the recruitment of

neutrophils and disproportionate cytokine production are strongly regulated to dampen inflammation and reduce collateral damage via neutrophil-derived mediators (Aragon-Vela et al., 2021; Scotland et al., 2011). On the other hand, males prioritize the activation of a robust immune response, heightened debris clearance and cell proliferation during this same timeframe. This suggests

a more aggressive repair strategy at the risk of driving an excessive inflammatory reaction. As for females, this is similar to what others have reported on the male response to musculoskeletal tissue damage (Aragon-Vela et al., 2021). By 72 h post-injury, females continue to prioritize modulation of the immune response, particularly as they upregulate cell necrosis and begin activating

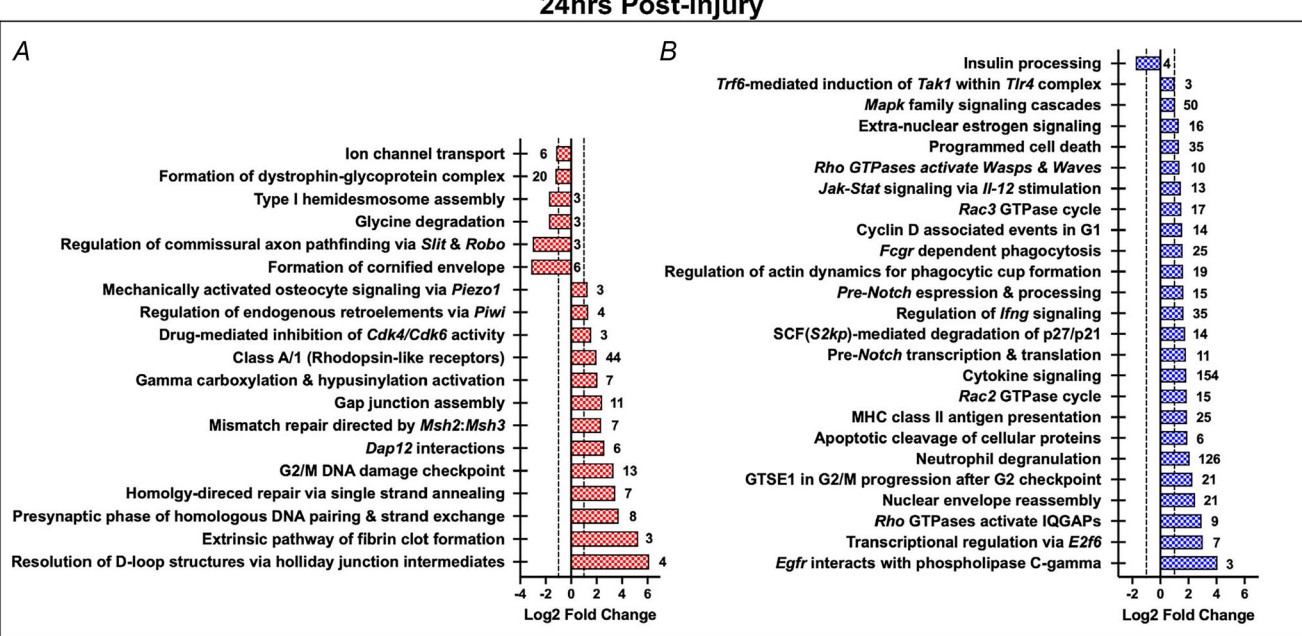

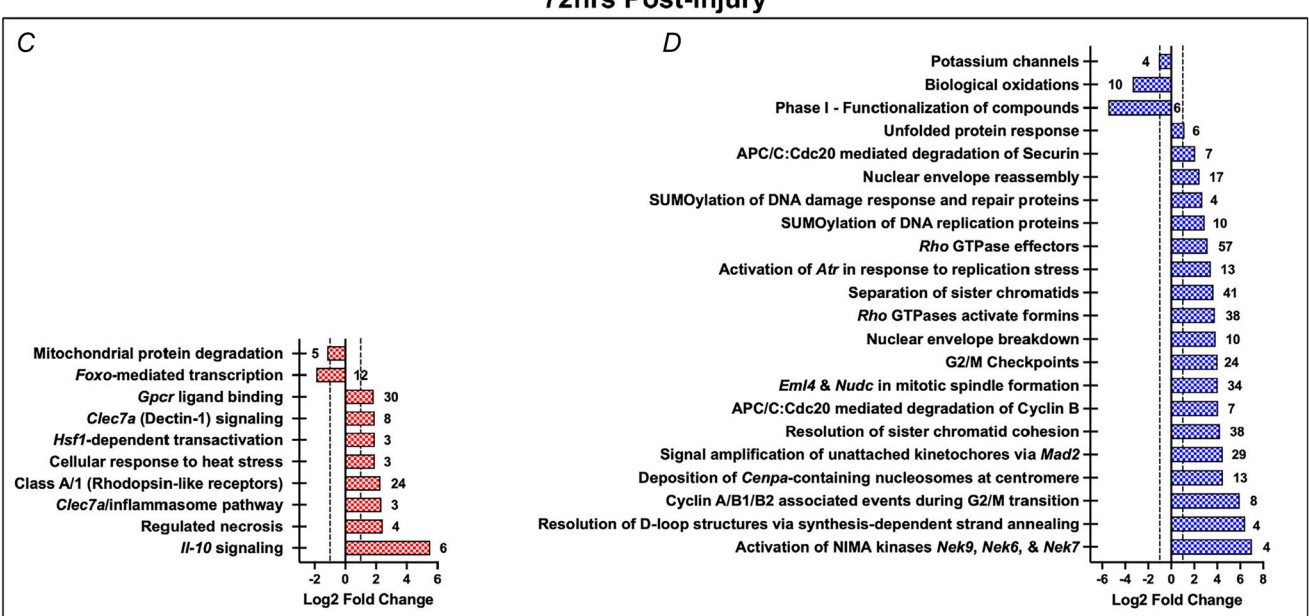

**Figure 9. Unique enriched biomolecular pathways**
Unique enriched biomolecular pathways with number of associated genes (≥ 3 genes; $q < 0.05$; log2 fold change > 1). *A*, females 24 h post-injury. *B*, males 24 h post-injury. *C*, females 72 h post-injury. *D*, males 72 h post-injury.

inflammatory cascades via *Clec7a* to recruit immune cells and clear debris. By contrast, males largely shift away from the inflammatory response and emphasize reparative processes through enhanced cell proliferation, cytoskeletal reorganization and protein stability.

Taken as a whole, the sex differences in the early response to ACL overuse suggests females are methodically taking steps that support sustained and regulated healing, whereas males are taking a more aggressive approach to accelerate tissue repair. Both strategies come with potential consequences. The more methodical approach taken by females may ultimately result in a more stable reparative process, although it may take them longer to restore ACL structure and function. The more aggressive approach taken by males may enable a faster reparative process, but may increase the risk of excessive fibrotic activity and scar formation (Ashcroft & Mills, 2002). However, in the context of sport, these sex-specific reparative strategies may in part explain the female–male disparity in ACL overuse injury and failure rates. If female ACLs accrue a greater volume of matrix damage and have a slower reparative response, then high-intensity reloading of the ACL prior to resolution of the injury could lead to further accumulation and propagation of matrix damage across collagen length scales (i.e. triple helix, fibril and fibre) (Chen et al., 2019; Kim et al., 2022; Loflin et al., 2023; Putera et al., 2023). By contrast, a faster reparative response in male ACLs could facilitate a quicker reengagement of high-intensity activity at the same time as mitigating the immediate risk of a recurrent submaximal injury. If confirmed, this would warrant greater consideration for sex-specific training and recovery regimens in load management. There is a current focus on enhancing athletic performance by emphasizing active management of loads experienced during training and conditioning (Buser et al., 2025; Ferraz et al., 2025; Leupold et al., 2025; Zaremski et al., 2025). A similar approach could be critical in preventing ligament overuse and subsequent tissue failures. Implementation of sex-specific training and recovery strategies following an overuse injury may be necessary for effective ligament repair and the mitigation of excessive ACL wear and tear (i.e. matrix damage) (Nyland, 2023; Nyland et al., 2022).

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

## Additional information

### Data availability statement

All mechanical, histological and molecular data supporting the study results are reported in the paper itself and the associated Supporting Information.

### Competing interests

EMW is Editor-in-Chief of *Sports Health: A Multidisciplinary Approach* peer-reviewed journal and the Chair of the Clinical Advisory Board for ProTherapeutics. No other authors have anything to disclose.

### Author contributions

S.H.S. was responsible for conceptualization, data curation, investigation, formal analysis, funding acquisition, methodology, project administration, resources, software, supervision, validation, writing – original draft, writing – reviewing & editing. B.E.L. was responsible for conceptualization, data curation, investigation, formal analysis, software, supervision, validation, writing – reviewing & editing. A.R.C. was responsible for investigation, formal analysis, software, writing – reviewing & editing. R.H. was responsible for data curation, investigation, formal analysis, software, writing – reviewing & editing. S.S. was responsible for data curation, formal analysis, writing – reviewing & editing. E.M.W.

was responsible for conceptualization, formal analysis, writing – original draft, writing – reviewing & editing. All authors approved the final version of the manuscript submitted for publication and agree to be accountable for all aspects of the work. All persons designated as authors qualify for authorship and all those who qualify are listed.

## Funding

Support for this study includes NIH/NIAMS funding (SHS, AR070903).

## Acknowledgements

We thank the Indiana Center for Musculoskeletal Health Histology Core for facilitating tissue processing and the Indiana University School of Medicine Center for Medical Genomics and Bioinformatics for their assistance with the RNA sequencing experiment.

## Keywords

anterior cruciate ligament, fatigue loading, mouse model, sex differences, tissue overuse

## Supporting information

Additional supporting information can be found online in the Supporting Information section at the end of the HTML view of the article. Supporting information files available:

**Peer Review History**
**Supplemental File 1**
**Supplemental File 2**
**Supplemental File 3**

