## [Peer Review History · The Journal of Physiology]

Sex differences in the physiological response to acute anterior cruciate ligament overuse

Stephen H Schlecht, Ben E Loflin, Adam R Carter, Roufael Hanna, Simran K Shergill, and Edward M Wojtys
DOI: 10.1113/JP289078

Corresponding author(s): Stephen Schlecht (steschle@iu.edu)

The following individual(s) involved in review of this submission have agreed to reveal their identity: Christopher S Fry (Referee #1)

Review Timeline:

Submission Date:	29-Apr-2025
Editorial Decision:	12-Aug-2025
Revision Received:	03-Sep-2025
Accepted:	23-Sep-2025

Senior Editor: Paul Greenhaff

Reviewing Editor: Martino Franchi

Transaction Report:

Dear Dr Schlecht,

Re: JP-RP-2025-289078 "**Sex differences in the physiological response to acute anterior cruciate ligament overuse**" by Stephen H Schlecht, Ben E Loflin, Adam R Carter, Roufael Hanna, Simran K Shergill, and Edward M Wojtys

Thank you for submitting your manuscript to The Journal of Physiology. It has been assessed by a Reviewing Editor and by 1 expert referees and we are pleased to tell you that it is potentially acceptable for publication following satisfactory major revision.

REVISION CHECKLIST:

Please upload two versions of your manuscript text: one with all relevant changes highlighted and one clean version with no

changes tracked. The manuscript file should include all tables and figure legends, but each figure/graph should be uploaded as separate, high-resolution files.

We look forward to receiving your revised submission.

Yours sincerely,

Paul Greenhaff
Senior Editor
The Journal of Physiology

REQUIRED ITEMS

1) - Author photo and profile. First or joint first authors are asked to provide a short biography (no more than 100 words for one author or 150 words in total for joint first authors) and a portrait photograph. These should be uploaded and clearly labelled together in a Word document with the revised version of the manuscript. See Information for Authors for further details.

2) - You must start the Methods section with a paragraph headed Ethical approval (https://jp.msubmit.net/cgi-bin/main.plex?form_type=display_requirements#methods).

Research must comply with The Journal's policies regarding animal experiments (<https://physoc.onlinelibrary.wiley.com/hub/animal-experiments>) and adherence to these policies must be stated in the manuscript.

Authors should confirm in their Methods section that their experiments were carried out according to the guidelines laid down by their institution's animal welfare committee, including an ethics approval reference number. The Methods section must contain a statement about access to food, water and housing, details of the anaesthetic regime: anaesthetic used, dose and route of administration, and method of killing the experimental animals.

3) - Your manuscript must include a complete Additional Information section, including competing interests; funding; author contributions and acknowledgements.

4) - Please upload separate high-quality figure files via the submission form.

5) - Please ensure that any tables are editable and in Word format, and wherever possible, embedded in the article file itself.

6) - Please ensure that the Article File you upload is a Word file.

7) - Papers must comply with the Statistics Policy: https://jp.msubmit.net/cgi-bin/main.plex?form_type=display_requirements#statistics.

In summary:

- If n {less than or equal to} 30, all data points must be plotted in the figure in a way that reveals their range and distribution. A bar graph with data points overlaid, a box and whisker plot or a violin plot (preferably with data points included) are acceptable formats.

- If $n > 30$, then the entire raw dataset must be made available either as supporting information, or hosted on a not-for-profit repository, e.g. FigShare, with access details provided in the manuscript.

- 'n' clearly defined (e.g. x cells from y slices in z animals) in the Methods. Authors should be mindful of pseudoreplication.
- All relevant 'n' values must be clearly stated in the main text, figures and tables.
- The most appropriate summary statistic (e.g. mean or median and standard deviation) must be used. Standard Error of the Mean (SEM) alone is not permitted.
- Exact p values must be stated. Authors must not use 'greater than' or 'less than'. Exact p values must be stated to three significant figures even when 'no statistical significance' is claimed.

8) - A Data Availability Statement is required for all papers reporting original data. This must be in the Additional Information section of the manuscript itself. It must have the paragraph heading 'Data Availability Statement'. All data supporting the results in the paper must be either: in the paper itself; uploaded as Supporting Information for Online Publication; or archived in an appropriate public repository. The statement needs to describe the availability or the absence of shared data. Authors must include in their statement: a link to the repository they have used, or a statement that it is available as Supporting Information; reference the data in the appropriate sections(s) of their manuscript; and cite the data they have shared in the References section. Whenever possible, the scripts and other artefacts used to generate the analyses presented in the paper should also be publicly archived. If sharing data compromises ethical standards or legal requirements then authors are not expected to share it, but must note this in their statement. For more information, see our Statistics Policy.

9) - Please include an Abstract Figure file, as well as the Figure Legend text within the main article file. The Abstract Figure is a piece of artwork designed to give readers an immediate understanding of the research and should summarise the main conclusions. If possible, the image should be easily 'readable' from left to right or top to bottom. It should show the physiological relevance of the manuscript so readers can assess the importance and content of its findings. Abstract Figures should not merely recapitulate other figures in the manuscript. Please try to keep the diagram as simple as possible and without superfluous information that may distract from the main conclusion(s). Abstract Figures must be provided by authors no later than the revised manuscript stage and should be uploaded as a separate file during online submission labelled as File Type 'Abstract Figure'. Please also ensure that you include the figure legend in the main article file. All Abstract Figures should be created using BioRender. Authors should use The Journal's premium BioRender account to export high-resolution images. Details on how to use and access the premium account are included as part of this email.

10) - The corresponding author must provide an institutional email address (not a personal address) for their author account and state this email within the manuscript. We encourage ALL co-authors to also provide institutional email addresses. If this cannot be provided (as corresponding author), then a stamped letter must be provided from the institution which confirms their role and employment there (please upload this with the revised submission).

EDITOR COMMENTS

Reviewing Editor:

Dear Authors,

Thank you for submitting your research to The Journal of Physiology. Your article has been carefully reviewed by an expert referee and a Reviewing Editor. I am pleased to let you know that we both found the topic of great interest as the burden of ACL injury is much higher in females. The data provided yield novel insight into dysregulated processes unique to each sex.

One of the major comments that I share with the Referee is that the integration of the RNA-seq datasets between sexes at each time point would be a crucial addition as that will much more strongly allow authors to parse out sex-specific signatures vs the individualised genes lists and ontology analysis that is currently presented. In this way the impact of the RNA-seq findings would be dramatically strengthened as well as be far more rigorous than the current qualitative comparison. I strongly recommend the authors to consider all Referee's comments, but this one in particular too.

Senior Editor:

Thank you for the manuscript submission and apologies on the delay in review, which was due to one reviewer letting us down. Nevertheless, we have comments from a reviewing editor and a specialist referee. Both can see merit in the work, but the Reviewing Editor has provided a series of comments that the authors need to consider, particularly the integration of the RNA-seq datasets between sexes at each time point. Please also consult the Journal of Physiology requirements for reporting in animal experiments: <https://physoc.onlinelibrary.wiley.com/hub/animal-experiments>

Please could you confirm in the section "Statistical Analysis" that the data depicted in the figures refers to mean +/- standard deviation unless otherwise stated.

Please note Journal policy mandates, rather than recommends, some of the elements of the guidance. We look forward to receiving the revised manuscript.

REFeree COMMENTS

Referee #1:

Schlecht and colleagues present findings from a pre-clinical study exploring the biological effect of sex on anterior cruciate ligament (ACL) morphology and recovery following overuse injury. Ligament injuries result in robust and persistent functional deficits that remain despite surgical reconstruction and targeted rehabilitation. The burden of musculoskeletal injury is much higher in females, who also have poorer recovery and long-term outcomes in addition to greater re-injury rates. Elucidating sex-specific effectors of injury risk and recovery would allow for the development of stronger evidence-based prevention strategies. The study conducted is robust with consideration given to molecular and morphological outcomes at two separate time points post-injury. The manuscript is well written and the conclusions supported by the results. I have a few comments in an effort to further strengthen an already strong manuscript:

Can the authors report body weight for the 2 sexes? And this is likely a bit of ignorance on my part, but are values for max force and similar parameters expressed per unit bodyweight? I acknowledge I am a muscle biologist, and often to more directly compare strength differences between males and females (human or murine), force is often presented as a bodyweight normalized value given the size difference. I am curious if that is common for ligament failure loads, and even if not, I would be interested in how the max force values presented in Figure 2 would compare between the sexes.

The longitudinal RNA-seq data are interesting. The author's conclusions could be further strengthened by integrating the male and female RNA-seq datasets. I was expecting to encounter gene lists or even Venn diagrams showing shared and unique genes at each time point between females and males (shared up-regulated genes, shared down-regulated genes, sex-specific up-regulated genes, etc.). These would be very simple quantitative data to visually represent how similar or dissimilar the transcriptomic response is between sexes. The authors would have an easy time to simply identify the percent overlap in differential genes between species after injury. Following dataset integration, I would be very interested in a principle component analysis for each time point (Male Control, Male Fatigue, Female Control, Female Fatigue) as a robust visual tool to further underscore a sex-specific global shift in the ligament transcriptome. Ontological analyses could then be conducted on the sex-specific gene set lists to further parse out sex-unique adaptation.

I could continue to parse out the RNA-seq data for days, and I want to limit comments on specific genes, but the acute upregulation of periostin in the female fatigued ACL's jumped out at me given its association with posttraumatic osteoarthritis following joint injury. Do the authors see a similar induction of periostin in the males?

END OF COMMENTS

We thank the reviewers for their excellent comments. We feel we have addressed all of them in the revised manuscript. We hope the reviewers agree that this is a much better paper now. In addition to the reviewer comments, we have also made manuscript edits and added new documents (e.g., bio) to comply with JP's publishing standards. For transparency, we have also provided two copies of the manuscript revision, with one highlighting changes made.

Reviewing Editor:

One of the major comment that I share with the Referee is that the integration of the RNA-seq datasets between sexes at each time point would be a crucial addition as that will much more strongly allow authors to parse out sex-specific signatures vs the individualized genes lists and ontology analysis that is currently presented. In this way the impact of the RNA-seq findings would be dramatically strengthened as well as be far more rigorous than the current qualitative comparison. I strongly recommend the authors to consider all Referee's comments, but this one in particular too.

Thank you for these comments. We agree with both reviewers on this point. We have now included 4-way Venn diagrams in Figures 6 and 7 to exhibit shared and unique genes. Additionally, we performed PCA on control and fatigued ACL DEGs at each timepoint and provided the PC1 – PC2 plots in Figure 8. We also added text in the methods and results. All PCA data are available in Supplemental File 2. We also investigated Gene Ontology and highlighted the terms that are unique to males and females at both post-injury timepoints in Tables 1 and 2, as well as added text in the methods and results. All GO data (shared and unique terms) are available in Supplemental File 3. As for the Reactome-derived pathways, these are the result of integrated datasets after down-weighting. To maintain consistency with reporting of unique GO terms, we revised Figure 9 to highlight all significantly enriched pathways not shared between males and females (i.e. unique). The full pathways list for males and females that met cutoff criteria (≥ 3 genes; $q < 0.05$; \log_2 fold change > 1) are provided in Supplemental File 4.

Referee #1:

Schlecht and colleagues present findings from a pre-clinical study exploring the biological effect of sex on anterior cruciate ligament (ACL) morphology and recovery following overuse injury. Ligament injuries result in robust and persistent functional deficits that remain despite surgical reconstruction and targeted rehabilitation. The burden of musculoskeletal injury is much higher in females, who also have poorer recovery and long-term outcomes in addition to greater re-injury rates. Elucidating sex specific effectors of injury risk and recovery would allow for the development of stronger evidence-based prevention strategies. The study conducted is robust with consideration given to molecular and morphological outcomes at two separate time points post-injury. The manuscript is well written and the conclusions are supported by the results. I have a few comments in an effort to further strengthen an already strong manuscript: Can the authors report body weight for the 2 sexes? And this is likely a bit of ignorance on my part, but are values for max force and similar parameters expressed per unit bodyweight? I acknowledge I am a muscle biologist, and often to more directly compare strength differences between males and females (human or murine), force is often

presented as a bodyweight normalized value given the size difference. I am curious if that is common for ligament failure loads, and even if not, I would be interested in how the max force values presented in Figure 2 would compare between the sexes. The longitudinal RNA-seq data are interesting. The author's conclusions could be further strengthened by integrating the male and female RNA-seq datasets. I was expecting to encounter gene lists or even Venn diagrams showing shared and unique genes at each time point between females and males (shared up-regulated genes, shared down-regulated genes, sex specific up-regulated genes, etc.). These would be very simple quantitative data to visually represent how similar or dissimilar the transcriptomic response is between sexes. The authors would have an easy time to simply identify the percent overlap in differential genes between species after injury. Following dataset integration, I would be very interested in a principal component analysis for each time point (Male Control, Male Fatigue, Female Control, Female Fatigue) as a robust visual tool to further underscore a sex-specific global shift in the ligament transcriptome. Ontological analyses could then be conducted on the sex specific gene set lists to further parse out sex-unique adaptation. I could continue to parse out the RNA-seq data for days, and I want to limit comments on specific genes, but the acute upregulation of periostin in the female fatigued ACLs jumped out at me given its association with posttraumatic osteoarthritis following joint injury. Do the authors see a similar induction of periostin in the males?

Thank you very much for these comments. Our response to the Reviewing Editor addresses many of these comments. With regards to body weights, we have provided the mean and standard deviations of males and females in the captions of Figures 2 and 3. All mechanical data is adjusted by body weight via linear regression prior to statistical analysis, thus the data is not reported per unit of body weight. We have previously adjusted mechanical measures by ACL cross-sectional area or fat mass/lean mass, with no significant differences between outcomes.

Dear Dr Schlecht,

Re: JP-RP-2025-289078R1 "**Sex differences in the physiological response to acute anterior cruciate ligament overuse**" by Stephen H Schlecht, Ben E Loflin, Adam R Carter, Roufael Hanna, Simran K Shergill, and Edward M Wojtys

We are pleased to tell you that your paper has been accepted for publication in The Journal of Physiology.

Yours sincerely,

Paul Greenhaff
Senior Editor
The Journal of Physiology

If you would like to receive our 'Research Roundup', a monthly newsletter highlighting the cutting-edge research published in The Physiological Society's family of journals (The Journal of Physiology, Experimental Physiology, Physiological Reports, The Journal of Nutritional Physiology and The Journal of Precision Medicine: Health and Disease), please click this link, fill in your name and email address and select 'Research Roundup':
<https://www.physoc.org/journals-and-media/membernews>

- **TRANSPARENT PEER REVIEW POLICY:** To improve the transparency of its peer review process, The Journal of Physiology publishes online as supporting information the peer review history of all articles accepted for publication. Readers will have access to decision letters, including Editors' comments and referee reports, for each version of the manuscript as well as any author responses to peer review comments. Referees can decide whether or not they wish to be named on the peer review history document.
- You can help your research get the attention it deserves! Check out Wiley's free Promotion Guide for best-practice recommendations for promoting your work at: www.wileyauthors.com/eoo/guide. You can learn more about Wiley Editing Services which offers professional video, design, and writing services to create shareable video abstracts, infographics, conference posters, lay summaries, and research news stories for your research at: www.wileyauthors.com/eoo/promotion.
- **IMPORTANT NOTICE ABOUT OPEN ACCESS:** To assist authors whose funding agencies mandate public access to published research findings sooner than 12 months after publication, The Journal of Physiology allows authors to pay an Open Access (OA) fee to have their papers made freely available immediately on publication.

EDITOR COMMENTS

Reviewing Editor:

The authors have done a great job in providing additional analyses as asked by the reviewer.

These data provide unique insight underscoring the unique biological response to strain injury between sexes.

Senior Editor:

Thank you for addressing the comments previously raised by the Reviewing Editor and Reviewer, which included the integration of the RNA-seq datasets to allow additional insight into sex-specific transcriptomic responses. Both the Reviewing Editor and Reviewer feel the authors have done a good job at revising the manuscript, which is now deemed suitable for publication. Thank you for considering The Journal of Physiology to publish your research.

REFeree COMMENTS

Referee #1:

The authors have addressed my comments from the initial submission. The integration of the RNA-seq datasets provides additional insight into sex-specific transcriptomic responses to strain.